# Volatile Sulfur Compounds Produced by the Anaerobic Bacteria *Porphyromonas* spp. Isolated from the Oral Cavities of Dogs

**DOI:** 10.3390/vetsci10080503

**Published:** 2023-08-04

**Authors:** Noriyuki Ito, Naoyuki Itoh, Satoshi Kameshima

**Affiliations:** Laboratory of Small Animal Internal Medicine, School of Veterinary Medicine, Kitasato University, Towada 034-8628, Japan; dv15002s@st.kitasato-u.ac.jp (N.I.); skame@vmas.kitasato-u.ac.jp (S.K.)

**Keywords:** volatile sulfur compounds (VSCs), *Porphyromonas gulae*, dog, periodontal disease, halitosis

## Abstract

**Simple Summary:**

Periodontal disease is the most common oral disease in dogs, and their owners usually notice this disorder in their animals via halitosis. The volatile sulfur compounds (VSCs), such as hydrogen sulfide (H_2_S), methyl mercaptan (CH_3_SH), and dimethyl sulfide ((CH_3_)_2_S), produced by periodontal-disease-associated bacteria cause halitosis. In addition, VSCs are not only the cause of halitosis, but are also toxic to oral tissues. *Porphyromonas gulae* is the main pathogenic bacterium in canine periodontal disease. However, the characteristics of the VSCs produced by *P. gulae* are unknown. The results of the present study demonstrate that *P. gulae* and other *Porphyromonas* spp. that were isolated from the oral cavities of dogs with periodontal disease can produce large amounts of H_2_S and CH_3_SH. In particular, the amount of CH_3_SH was dominant. It was suggested that the high levels of H_2_S and CH_3_SH in *P. gulae* and other *Porphyromonas* spp. contribute to halitosis and the destruction of periodontal tissues during the progression of periodontal disease in dogs. Appropriate oral care is required to prevent halitosis and the toxic effects of VSCs in dogs with periodontal disease.

**Abstract:**

*Porphyromonas* spp. are oral anaerobic Gram-negative bacteria that form black-pigmented colonies on blood agar and produce volatile sulfur compounds (VSCs), such as hydrogen sulfide (H_2_S), methyl mercaptan (CH_3_SH), and dimethyl sulfide ((CH_3_)_2_S), which cause halitosis and the destruction of periodontal tissues. *P. gulae* is considered the main pathogen involved in periodontal disease in dogs. However, the characteristics of the VSCs produced by *P. gulae* are unknown. In the present study, VSCs were measured in 26 isolates of *P. gulae* and some isolates of the other *Porphyromonas* spp. obtained from the oral cavities of dogs with periodontal disease using an in vitro assay with an Oral Chroma^TM^ gas chromatograph. The results demonstrated that *P. gulae* was able to produce large amounts of H_2_S and CH_3_SH, and the dominant product was CH_3_SH (CH_3_SH/H_2_S was approximately 2.2). Other *Porphyromonas* spp. that were also obtained from the oral cavities of dogs with periodontal disease indicated similar levels of production of H_2_S and CH_3_SH to those of *P. gulae*. It is strongly suggested that the high levels of H_2_S and CH_3_SH produced by *P. gulae* and other *Porphyromonas* spp. contribute to halitosis and the destruction of periodontal tissues during the progression of periodontal disease in dogs.

## 1. Introduction

Volatile sulfur compounds (VSCs) are the main cause of halitosis (malodor) in humans and dogs with periodontal disease [1,2,3,4]. The major components of VSCs are hydrogen sulfide (H_2_S), methyl mercaptan (CH_3_SH), and dimethyl sulfide ((CH_3_)_2_S), which are derived from sulfur-containing amino acids (e.g., l-cysteine and l-methionine) [2,5,6]. Oral anaerobic Gram-negative bacteria that form black-pigmented colonies on blood agar, such as *Porphyromonas* spp., contribute significantly to the production of VSCs and are major pathogens in periodontal disease [2,4,6,7,8,9]. Simultaneously, it has also been suggested that, even at low levels, VSCs have cytotoxic potential for oral tissues and are associated with the progress of periodontal disease [5,10]. Specifically, the levels of VSC production are likely to reflect the pathogenicity of bacteria. In addition, since VSCs can be transported to various organs of the body via the blood and inhibit the normal function and metabolic activity of tissues and organs, the development of systemic diseases is a concern [11].

In dogs, periodontal disease is the most common disease, and a high prevalence (44–100%) has been reported [12]. Canine periodontal disease increases with age but is predominant in toy breeds of dogs from the younger stages of their lives [12,13]. In periodontal disease, tooth loss is induced by destruction of the soft and hard tissues supporting tooth structures in final stage [14,15]. In addition, as periodontal disease often follows pain, the dogs cannot maintain its quality of life. Recently, in Japan, elderly dogs and toy breeds of dogs have been growing in number. Halitosis is the first chance for owners to notice the presence of periodontal disease in their companions. However, due to owners’ frequent and close contact with dogs, halitosis has also developed into a serious problem that causes anguish for owners.

The VSCs produced by human periodontal-disease-associated bacteria were assessed in vitro [16,17,18]. In dogs, in vivo analyses of VSCs were performed in clinical cases with or without periodontal disease [3,4,19]. However, there are no available reports regarding the in vitro assessment of VSC production via periodontal-disease-associated *Porphyromonas* spp. in dogs. Therefore, the present study aimed to evaluate the characteristics of the VSCs produced by *Porphyromonas* spp. (*Porphyromonas gulae*, *Porphyromonas macacae*, *Porphyromonas gingivalis*, and *Porphyromonas gingivicanis*) isolated from canine oral cavities using an in vitro assay. Simultaneously, for comparison with *Porphyromonas* spp., we measured the VSCs produced by *Bacteroides pyogenes*, which is also an anaerobic Gram-negative bacterium, but which does not form black-pigmented colonies on blood agar; it was isolated from canine oral cavities containing commensal microbiota or periodontal-disease-associated bacteria [20,21,22].

## 2. Materials and Methods

### 2.1. Ethics Statements and Sample Collection

The present study was approved by the ethics committees of two facilities: the Veterinary Teaching Hospital of Kitasato University (approval number: 2022-8001) and the Ito Animal Hospital (approval number: 1122-12-001). A total of 46 dogs with diagnosed gingivitis were included in the study. The owners noticed the various grades of dental plaque in their dogs and desired the assessment of the gingival status of their animals as a part of routine physical examinations, without sedation and/or anesthesia. Due to non-use of sedation and/or anesthesia, the probing of the gingival sulcus was not performed. Therefore, we assessed the gingival status using the original simplified index, modified from a previous study [23], via gingival macroscopic findings. The gingival condition was classified into four stages (gingival index 0 = normal gingiva, 1 = slight redness and edema, 2 = gingival is red and swollen, 3 = gingiva is red or reddish blue with swollen margins and a tendency to hemorrhage). The gingival indexes in the present 46 dogs were all above index 1. After informed consent, the owners decided whether or not to accept and to permit the submission of oral swab samples from their animals of their own free will.

The investigated 46 dogs constituted 25 males and 21 females, aged 2–19 years old (mean ± standard deviation: 8.6 ± 4.8, median: 8.0), and 20 breeds including Toy Poodle (*n* = 2), Pug (*n* = 1), Labrador Retriever (*n* = 3), Kuvasz (*n* = 1), Golden Retriever (*n* = 3), Pomeranian (*n* = 3), Yorkshire Terrier (*n* = 2), Miniature Dachshund (*n* = 6), Cavalier King Charles Spaniel (*n* = 2), Tibetan Spaniel (*n* = 1), Maltese (*n* = 1), Mongrel (*n* = 7), Japanese Chin (*n* = 1), Shiba Dog (*n* = 1), Shetland Sheepdog (*n* = 1), Jack Russell Terrier (*n* = 1), Chihuahua (*n* = 2), Bulldog (*n* = 1), Pembroke Welsh Corgi (*n* = 1), and Miniature Schnauzer (*n* = 1). The body weights of the dogs ranged from 2.4–37.8 kg (mean ± standard deviation: 10.7 ± 10.6, median: 6.9), and their stages of gingivitis (= gingival index) were 1.7 ± 0.8 with median of 1.0 (index 1: *n* = 25, index 2: *n* = 12, index 3: *n* = 9).

To detect the pathogenic microorganisms in the dogs, oral swab specimens were collected at the position of the gingival margin from the teeth of the canine to fourth premolars on the left side of the maxilla in dogs without treatment with sedation and/or anesthesia by using a sterilized device (TX709A, Clean Foam^®^ Series, Texwipe, Kernersville, NC, USA).

### 2.2. Bacterial Culture and Isolation

The obtained swab samples were immediately inoculated on a non-selective anaerobic bacterial isolation agar plate that included hemin, vitamin K, l-arginine, and hemolytic rabbit blood (ABHK Agar Plate; Nissui Pharmaceutical Co., Ltd., Tokyo, Japan). The culture plates were incubated at 37 °C for 10 days under aerobic conditions using a commercial, disposable, self-contained anaerobic system (ANAEROMATE^®^-P “Nissui”, Nissui Pharmaceutical Co., Ltd., Tokyo, Japan). Then, black-pigmented bacterial colonies from 42 dogs and non-black-pigmented colonies from 4 other dogs were randomly selected and pure-cultured for 5 days according to the same procedure.

### 2.3. Bacterial Production of VSCs and Determination of VSCs

The concentrations of suspensions of pure-cultured bacterial colonies in axenic phosphate-buffered saline (pH 7.4) were adjusted to have an optical density of 2.0 ± 0.1 at 600 nm by using SimpliNano (GE Health Care Ltd., Chicago, IL, USA). To determine the VSCs, the prepared 250 µL suspensions were placed in a KBM Anaerobic Test Tube Medium (Kohjin Bio Co., Ltd., Saitama, Japan) that contained 15 mL of liquid medium, which included enough nutrients for a wide range of anaerobic bacteria; this was incubated at 37 °C for 72 h under anaerobic conditions. Prior to the inoculation of the bacterial suspension into the culture bottle, a small hole was made in the screw cap of the bottle via puncturing and was sealed with a butyl rubber adhesive tape. After 72 h of incubation, the produced gas was collected from the upper layer of the air in the culture bottle using a disposable syringe with a 26G needle that was inserted through the small hole in the screw cap that was sealed with tape. The levels of VSCs (H_2_S, CH_3_SH, and (CH_3_)_2_S) were measured using an Oral Chroma^TM^ gas chromatograph (CHM-2, FIS Inc., Hyogo, Japan). In addition, the ratio of H_2_S to CH_3_SH (CH_3_SH/H_2_S) was calculated.

### 2.4. Molecular Identification of Bacterial Species

For the molecular identification of the bacterial species, the 200 µL suspensions used for VSC determination were also used to extract the total bacterial DNA using the commercial kit (NucleoSpinl^®^ DNA Stool, MACHEREY-NAGEL, GmbH & Co. KG, Düren, Germany) according to the manufacturer’s instructions. The obtained DNA specimens were stored at −20 °C prior to the analysis.

According to the attached manual, fast PCR targeting 16S rRNA genes was performed using a commercial kit (Bacterial 16S rDNA PCR Kit Fast (800), Takara Bio Inc., Shiga, Japan); this included the universal primer pairs (10F: GTTTGATCCTGGCTCA; 800R: TACCAGGGTATCTAATCC) in order to amplify regions of approximately 800 bp, a premix (the master mix was composed of modified Taq DNA polymerase and dNTPs), and a positive control. In brief, the total reaction volume was set to 25.0 µL, which included 12.5 µL of the master mix, 1.25 µL of each primer, 2.5 µL of the DNA template, and 7.5 µL of DNA-free water. The conditions of the fast PCR were as follows: 25 cycles of 92 °C for 5 s, 50 °C for 1 s, and 68 °C for 8 s. The amplicons used in the PCR were electrophoresed on a 1.5% agarose gel and were stained (AtlasSight DNA Stain, Bioatlas, Tartu, Estonia) for the visualization of the bands. The expected size of the DNA fragments was confirmed via transillumination under UV light.

The PCR products were purified using a commercial kit (NucleoSpinl^®^ Gel and PCR Clean-up, MACHEREY-NAGEL, GmbH and Co. KG, Düren, Germany). A sequencing analysis was executed in a commercial laboratory (FASMAC Co., Ltd., Atsugi, Kanagawa, Japan). Sequence alignment and compilation were performed using the MEGA 11.0.10 (https://www.megasoftware.net, accessed on 16 April 2023) program. To determine the bacterial species, the obtained DNA sequences were compared with GenBank references by using BLAST searches (http://www.ncbi.nlm.nih.gov/, accessed on 26 April 2023).

### 2.5. Data Analysis

The data were analyzed using “EZR” software (version 1.61). The measured values were expressed as the mean ± standard deviation. The Kruskal–Wallis test was used to compare the differences among groups, and the Steel–Dwass test was conducted to assess the differences between pairs of groups. Statistical differences were considered significant at *p* < 0.05.

## 3. Results

### 3.1. Molecular Identification of Bacteria

According to the molecular identification of the bacteria in the present study, among the 42 isolates of black-pigmented colony-forming bacteria were 26 isolates of *P. gulae*, 8 isolates of *P. macacae*, 4 isolates of *P. gingivalis*, and 4 isolates of *P. gingivicanis*. In addition, all four of the isolates of non-black-pigmented colony-forming bacteria were *Bacteroides pyogenes* (Table 1).

### 3.2. Volumes of VSCs Produced

The volumes of VSCs produced by the black-pigmented bacteria (*Porphyromonas* spp.) and non-black-pigmented bacteria (*B. pyogenes*) are shown in Table 2. Overall, the production of H_2_S and CH_3_SH in the *Porphyromonas* spp. group was revealed to have a higher tendency than that of *B. pyogenes*. Indeed, the volumes of H_2_S and CH_3_SH were significantly higher for *P. gulae* and *P. macacae* than for *B. pyogenes*. In addition, significant differences were observed between *P. gulae* and *P. macacae* regarding the levels of H_2_S and CH_3_SH. In other *Porphyromonas* spp., there were no statistical differences in H_2_S and CH_3_SH in comparison with *B. pyogenes* or among those species, which was due to the limited numbers that were investigated (e.g., there were only four isolates each of *P. gingivalis*, *P. gingivicanis*, and *B. pyogenes*) and the wide range of variation. The ratios of H_2_S to CH_3_SH (CH_3_SH/H_2_S) were approximately 2.2 for the *Porphyromonas* spp. group and 0.18 for *B. pyogenes*, and the ratios were significantly higher for *P. gulae* and *P. macacae* than for *B. pyogenes*. The volumes of (CH_3_)_2_S were irregular in all bacterial isolates, and many isolates were negative for the production of (CH_3_)_2_S (16/26 for *P. gulae*, 7/8 for *P. macacae*, 4/4 for *P. gingivalis*, 1/4 for *P. gingivicanis*, and 4/4 for *B. pyogenes*). No significant differences were demonstrated in the (CH_3_)_2_S levels among the investigated bacteria.

## 4. Discussion

The present study is the first report to demonstrate the characteristics of the VSCs produced by *P. gulae*, considered the main pathogen involved in periodontal disease in dogs [22,24], which was obtained from canine oral cavities. The results suggest that *P. gulae* has the potential to produce high levels of H_2_S and CH_3_SH, with CH_3_SH being dominant. As shown in the present study, the dominant gas was easy to determine due to the ratio of CH_3_SH/H_2_S. Although the examined numbers were insufficient, other *Porphyromonas* spp. (*P. macacae*, *P. gingivalis*, and *P. gingivicanis*) also indicated the same characteristics as those of *P. gulae* in terms of the VSCs that they produced. The values of H_2_S and CH_3_SH in *P. gulae* indicated positive kurtosis, suggesting that both VSCs’ production by *P. gulae* is stable and relatively low in deviation. In contrast, due to the *P. macacae* production of H_2_S and CH_3_SH showing negative kurtosis, the production of VSCs is likely to be unstable with wide variation. However, further investigation is needed to decide due to insufficient numbers in the present study. The characteristic of CH_3_SH-dominant VSC production was also reported in *P. gingivalis* in humans [16,17,18]. A clear difference was observed between *Porphyromonas* spp. and *B. pyogenes*. Regarding *B. pyogenes*, both H_2_S and CH_3_SH were produced at markedly low levels in comparison with *Porphyromonas* spp., and H_2_S was dominant in the VSCs, which was reflected in the low value of CH_3_SH/H_2_S. These differences in VSC levels and components between *Porphyromonas* spp. and *B. pyogenes* were easily calculable from the outset because these two bacteria belong to different genera. However, the enzyme l-cysteine desulfhydrase, which produces H_2_S from l-cysteine, was documented in both *Porphyromonas* spp. and *Bacteroides* spp. [2]. Additionally, l-methionine α-deamino-γ-mercaptomethane-lyase, which is an enzyme that produces CH_3_SH from l-methionine, was also demonstrated in both species of bacteria [2]. Therefore, it was suggested that the differences in the VSC levels and components between *Porphyromonas* spp. and *Bacteroides* spp. were derived from the levels of activity of these two enzymes. Despite being in the same genus, the VSCs produced by *P. gingivalis* and *P. endodontalis* had different components because *P. endodontalis* was shown to produce H_2_S-dominant VSCs in humans [16]. Due to the significant difference in the volumes of VSCs produced by *P. gulae* and *P. macacae*, it was also observed in the present study that the activities of l-cysteine desulfhydrase and l-methionine α-deamino-γ-mercaptomethane-lyase are likely to be characterized according to each species of bacteria.

Previous studies suggested that the main and important VSCs responsible for inducing halitosis are H_2_S and CH_3_SH, which are produced by some specific bacteria, such as *P. gingivalis*, *Fusobacterium nucleatum*, and *Tannerella forsythia* in humans [1,5,10]. The gases of H_2_S and CH_3_SH comprised approximately 90% of the VSCs in human breath [25], and some in vitro reports related to *P. gingivalis* in humans demonstrated a markedly low (almost negative) level of (CH_3_)_2_S production or even did not record it [16,17,18]. In addition, it is generally known that Gram-negative anaerobic bacteria predominantly produce H_2_S and CH_3_SH [1]. The importance of the H_2_S and CH_3_SH produced by periodontal-disease-associated bacteria was also confirmed in dogs through the present study. In the present study, although the details of the mechanism were unknown, the measured levels of (CH_3_)_2_S were unstable, and many samples revealed negative levels of (CH_3_)_2_S production.

The effects of VSCs are not only halitosis, but also toxicity, which is connected to the development of periodontal disease [5,10]. In practice, the VCSs of H_2_S and CH_3_SH play the main roles in injuries to periodontal tissues. In particular, it was indicated that CH_3_SH had the greatest harmfulness [5]. Therefore, the CH_3_SH/H_2_S ratio has the potential to represent toxic activity in tissues and the characteristics of each periodontal bacterium. In the present study, because large amounts of dominant CH_3_SH and non-dominant H_2_S were demonstrated in *P. gulae* and other *Porphyromonas* spp. from canine cases with periodontal disease, it was suggested that these bacteria induce severe damage to the periodontal tissues in animals and contribute to the progress of periodontal disease. In contrast, low levels of H_2_S and CH_3_SH suggest the weak pathogenicity of *B. pyogenes* in comparison with that of *Porphyromonas* spp. However, we cannot neglect this status in *B. pyogenes* because VCSs can be toxic to tissues even at low levels and with short-term exposure [2,5]. The free thiols (–SH groups) of both H_2_S and CH_3_SH can react with DNA and proteins [2,5,10]. The thiols penetrate deeply into gingival tissues [5,10]. Previous research has demonstrated that the mechanism of tissue damage by H_2_S and CH_3_SH was through the breaking of disulfide bonds in the proteins of the oral mucosa [5,10]. This is because disulfide bonds are critical for proteins’ structural integrity [5,10]. In addition, the inhibitory effect of CH_3_SH against proliferation and the apoptosis induced by H_2_S exposure were reported in oral epithelial cells [5,10]. Inhibited synthesis of collagen has been shown with CH_3_SH exposure [5,10]. Apoptosis and damage to DNA strands via H_2_S have also been demonstrated in human gingival fibroblasts [5,6,10]. Moreover, inflammatory reactions and immune responses, which included increased reactive oxygen species and inflammatory cytokine secretions, were initiated after damage to tissues and caused the degraded progress of periodontal disease [5,6,10]. Regarding (CH_3_)_2_S, this compound is essentially inert because it contains no reactive thiols [5].

## 5. Conclusions

The present study characterized the VSCs produced by *P. gulae* originating from canine periodontal disease. The results demonstrated that *P. gulae* can produce large amounts of H_2_S and CH_3_SH. In particular, CH_3_SH was revealed to be dominant (the CH_3_SH/H_2_S ratio was approximately 2.2). Other *Porphyromonas* spp. also indicated levels of production of H_2_S and CH_3_SH that were similar to those of *P. gulae*. The high levels of H_2_S and CH_3_SH contribute to halitosis and the destruction of periodontal tissues after inflammatory reactions and immune responses in relation to the progression of periodontal disease in dogs. In addition, H_2_S and CH_3_SH have the potential to be transported into other tissues and organs, and then inhibit their functions, resulting in the risk of systemic disease. Appropriate oral care is required to prevent the toxic effects of VSCs in dogs with periodontal disease.

## Figures and Tables

**Table 1 vetsci-10-00503-t001:** Molecular identification of the bacteria used in the present study.

Bacterial Species	Numbers Isolated	Reference Isolates in GenBank	Identity
*Porphyromonas gulae* (B)			
	1	JN713220	99.58%
	1	JN713221	99.86%
	1	JN713277	99.72%
	1	KM461998	99.58%
	9	KM462071	99.44–99.87%
	7	KM462153	99.44–99.86%
	3	LC749393	99.86–99.59%
	2	LC749394	99.59%, 99.86%
	1	LR134506	99.86%
*Porphyromonas macacae* (B)			
	2	AB547666	99.86%, 100%
	6	KM461959	99.58–99.86%
*Porphyromonas gingivalis* (B)			
	1	CP024594	99.72%
	1	CP024601	100%
	2	CP025931	99.86%, 100%
*Porphyromonas gingivicanis* (B)			
	3	NR_104833	99.46–99.72%
	1	JN713184	99.73%
*Bacteroides pyogenes* (NB)			
	1	HF558365	100%
	1	JN713205	100%
	1	MT271930	100%
	1	NR_041280	100%

B: black-pigmented colony formed; NB: non-black-pigmented colony formed.

**Table 2 vetsci-10-00503-t002:** The volumes of VSCs produced (mean ± standard deviation).

Bacterial Species	Number of Colonies	H_2_S (ppb)	CH_3_SH (ppb)	(CH_3_)_2_S (ppb)	CH_3_SH/H_2_S
Black-pigmented colony formed					
*Porphyromonas gulae*	26	6275.6 ± 341.6 ^(a)^	14,304.1 ± 1119.5 ^(d)^	64.2 ± 126.3	2.21 ± 0.15 ^(g)^
*Porphyromonas macacae*	8	5089.9 ± 973.3 ^(b)^	12,078.0 ± 1660.3 ^(e)^	0.9 ± 2.5	2.40 ± 0.17 ^(h)^
*Porphyromonas gingivalis*	4	6044.8 ± 485.5	13,471.5 ± 1578.5	0 ± 0	2.22 ± 0.14
*Porphyromonas gingivicanis*	4	6384.9 ± 408.1	14,292.3 ± 663.5	341.5 ± 327.1	2.24 ± 0.07
Non-black-pigmented colony formed					
*Bacteroides pyogenes*	4	2279.3 ± 1056.8 ^(c)^	505.1 ± 901.4 ^(f)^	0 ± 0	0.18 ± 0.32 ^(i)^

Significant differences. ^(a)^ vs. ^(b)^: *p* < 0.01, ^(a)^ vs. ^(c)^: *p* < 0.05, ^(b)^ vs. ^(c)^: *p* < 0.05, ^(d)^ vs. ^(e)^: *p* < 0.05, ^(d)^ vs. ^(f)^: *p* < 0.05, ^(e)^ vs. ^(f)^: *p* < 0.05, ^(g)^ vs. ^(i)^: *p* < 0.05, ^(h)^ vs. ^(i)^: *p* < 0.05. Additional data. ^(a)^ range: 5355.5–7079.0, median: 6540.5, kurtosis: 3.74. ^(b)^ range: 3537.0–6363.5, median: 5107.8, kurtosis: −1.03. ^(c)^ range: 1084.5–3442.5, median: 2295.0, kurtosis: -2.73. ^(d)^ range: 10,042.0–15714.0, median: 14,580.5, kurtosis: 7.78. ^(e)^ range: 9525.0–13,603.0, median: 12,456.3, kurtosis: −1.59. ^(f)^ range: 0–1852.0, median: 84.3, kurtosis: 3.84. ^(g)^ range: 1.68–2.45, median: 2.23, kurtosis: 5.93. ^(h)^ range: 2.14–2.69, median: 2.40, kurtosis: 0.43. ^(i)^ range: 0–0.65, median: 0.02, kurtosis: 3.89.

## Data Availability

The data presented in this study are included within the article.

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
