# Peer review of "Volatile Sulfur Compounds Produced by the Anaerobic Bacteria *Porphyromonas* spp. Isolated from the Oral Cavities of Dogs"

_vetsci, 2023, doi:10.3390/vetsci10080503_

Round 1
Reviewer 1 Report
The manuscript is well written, results were well presented but should add information about the prevalence of periodontal disease and problem associated with the disease
The language is good
Author Response
Reviewer 1
Comment sand Suggestions for Authors
The manuscript is well written, results were well presented but should add information about the prevalence of periodontal disease and problem associated with the disease
>
Thank you for your helpful advices.
In introduction, I have already provided the prevalence of periodontal disease in dogs.
And some sentence was described in introduction regarding the problem associated with the periodontal disease.
In addition, as the reviewer 3 also pointed out, the paragraph regarding systemic effects of periodontal disease in discussion was removed to introduction.

Reviewer 2 Report
Dear Authors,
the manuscript « Volatile Sulfur Compounds Produced by the Anaerobic Bacteria Porphyromonas spp. Isolated from the Oral Cavities of Dogs» by Noriyuki Ito, Naoyuki Itoh and Satoshi Kameshima (vetsci-2513181) is devoted to an interesting and important field at the “cross-section” of animal biochemistry, analytical chemistry, veterinary, animal diseases (such as gingivitis), etc. The author’s main aim was “to evaluate the characteristics of the VSCs produced by Porphyromonas spp. (Porphyromonas gulae, Porphyromonas macacae, Porphyromonas gingivalis, and Porphyromonas gingivicanis) isolated from canine oral cavities using an in vitro assay”. It is known that the volatile sulfur compounds (VSCs) are the following: hydrogen sulfide (H2S), methyl mercaptan (CH3SH), and dimethyl sulfide ((CH3)2S). This topic is an interesting and modern, taking into consideration the importance to measure such volatile sulfur compounds, “produced by periodontal-disease-associated bacteria cause halitosis”. The results demonstrated that “P. gulae can produce large amounts of H2S and CH3SH”. The authors mentioned that “CH3SH was revealed to be dominant (the CH3SH/H2S ratio was approximately 2.2)”. It is known that “the high levels of H2S and CH3SH contribute to halitosis and the destruction of periodontal tissues”.
I do not doubt the technical quality of the work and feel that there is a sufficient impact on a broader readership to justify publication in the "Veterinary Sciences".
There are some comments on the manuscript:
1. The most “serious” problem concerning this paper is the following: the sufficient amount of isolates for the detailed descriptive statistics is only shown for in the case of P. gulae (26 isolates). It is possible to accept (with some restrictions) the data for P. macacae (8 isolates), but not - for other colony-forming bacteria. It is necessary to provide the detailed descriptive statistics (standard error and deviation, min. and max. values, mean, mode, kurtosis, etc.) of the data, presented in the Table 2 (for P. gulae and P. macacae). This information can be discussed together with the obtained results in the context of the amounts of H2S and CH3SH
2. It is necessary to provide detailed information about the dogs in this study (breed or race, age, weight, etc.), as well as some information about their physiological and biochemical status (in particular, blood parameters that are usual for animal biochemistry and dog diagnostics in the veterinary clinics). It will be useful to calculate the correlation coefficients between VSCs and some biochemical blood parameters. This information must be discussed together with the obtained results in the context of the amounts of H2S and CH3SH (produced by P. gulae, etc.).
3. Table 2. There are some strange data obtained (64.2 ± 126.3 ppb, where std.error is twice higher as compared to the measured value) of the dimethyl sulfide ((CH3)2S) production by Porphyromonas gulae. The authors must clarify this matter in the text.
4. An interesting point concerning the low value of the CH3SH/H2S ratio (0.18) in the case of the four isolates of non-black-pigmented colony-forming bacteria Bacteroides pyogenes (Tables 1, 2) as compared to the data obtained for the black-pigmented colony-forming bacteria Porphyromonas spp. (“the CH3SH/H2S ratio was approximately 2.2”). This information must be discussed more fundamentally.
5. There are some misprints in English that I understand might be due to the point that English not being the first language of the authors.

Minor editing of English language required.
Author Response
Reviewer 2
Comments and Suggestions for Authors
There are some comments on the manuscript:
1.The most “serious” problem concerning this paper is the following: the sufficient amount of isolates for the detailed descriptive statistics is only shown for in the case of P. gulae (26isolates). It is possible to accept (with some restrictions) the data for P. macacae (8 isolates), but not - for other colony-forming bacteria. It is necessary to provide the detailed descriptive statistics 記述統計(standard error and deviation, min. and max. values, mean, mode, kurtosis尖度, etc.) of the data, presented in the Table 2 (for P. gulae and P. macacae). This information can be discussed together with the obtained results in the context of the amounts of H2S and CH3SH.
>
Thank you for advises.
I provided the additional data in P. gulae and P. macacae. Moreover, I discussed with H2S and CH3SH.
2. It is necessary to provide detailed information about the dogs in this study (breed or race, age, weight, etc.), as well as some information about their physiological and biochemical status (in particular, blood parameters that are usual for animal biochemistry and dog diagnostics in the veterinary clinics). It will be useful to calculate the correlation coefficients between VSCs and some biochemical blood parameters. This information must be discussed together with the obtained results in the context of the amounts of H2S and CH3SH (produced by P. gulae, etc.).
>
Thank you for suggestion.
I added some information regarding dogs.
I am sorry, I don’t have other data because this study was focused to determine the ability of VSCs production by P. gulae via in vitro assay using pure isolated colonies, not in vivo.
Moreover, as described in text, the owners were presented with a request of simple assessment of gingivitis in their dogs without sedation and/or anesthesia.
Of course, I understand your suggestion and we must investigate VSCs in dogs with some parameters in future.
3. Table 2. There are some strange data obtained (64.2± 126.3 ppb, where std. error is twice higher as compared to the measured value) of the dimethyl sulfide ((CH3)2S) production by Porphyromonas gulae. The authors must clarify this matter in the text.
>
Thank you for your indication.
I inserted one sentence in text.
4. An interesting point concerning the low value of theCH3SH/H2S ratio (0.18) in the case of the four isolates of non-black-pigmented colony-forming bacteria Bacteroides pyogenes (Tables 1, 2) as compared to the data obtained for the black-pigmented colony-forming bacteria Porphyromonas spp. (“theCH3SH/H2 S ratio was approximately 2.2”). This information must be discussed more fundamentally.
>
Thank you for advice.
I am very sorry, but we have already discussed this problem in discussion.
Due to there are no previous reports of VSCs produced by B. pyogenes, I cannot discuss any more.
5. There are some misprints in English that I understand might be due to the point that English not being the first language of the authors.
(/user/review/displayFile/40616764/T83C0DBj? file=review&report=30697549)
>
Thank you for your suggestion.
As I described in cover letter, this article has already reviewed by MDPI language service (English-Editing-Certificate-68107). But some sentences were added in this time. So, if it is needed, I agree with recheck of English.
Reviewer 3 Report
Dear authors,
Here is my contribution.
If animals only have gingivitis and not periodontitis, why not just address the problem in gingivitis?
Evoking periodontal disease should presuppose an evaluation (clinical and radiographic) and classification of the disease in the different animals used in the study.
Throughout the text: in vitro and in vivo italicized
1. Introduction
normal flora - commensal microbiota
2. Materials and Methods
2.1. Ethics Statements and Sample Collection
The authors report that animals have gingivitis, but there are varying degrees of gingivitis. Reference to this assessment (gingival index) should be provided.
2.3. Bacterial Production of VSCs and Determination of VSCs
The concentrations of suspensions of pure-cultured bacterial colonies in axenic PBS (please explain PBS meaning)
Discussion
"development of systemic diseases is a concern."
In my opinion the local expression and especially the systemic expression of periodontal disease, which is not just a concern but which is very well documented in man and dog, should be the subject of a paragraph in the introduction.
5. Conclusions
2.2) ?
References
Despite being old and the evaluation method being different, I believe that this reference should be taken into account:
Oral malodor measurements on a tooth surface of dogs with gingivitis. Hennet PR, Delille B, Davot JL.. Am J Vet Res. 1998 Mar;59(3):255-7.
On the other hand, this article may allow the authors to enrich the text with the explanation of the role of these compounds in the pathophysiology of periodontal disease.
Role of Hydrogen Sulfide in Oral Disease. Wu DD, Ngowi EE, Zhai YK, Wang YZ, Khan NH, Kombo AF, Khattak S, Li T, Ji XY. Oxid Med Cell Longev. 2022 Jan 25;2022:1886277. doi: 10.1155/2022/1886277. eCollection 2022.
Author Response
Reviewer 3
Comments and Suggestions for Authors
Dear authors,
Here is my contribution.
If animals only have gingivitis and not periodontitis, why not just address the problem in gingivitis?
>
Thank you for advice.
This study was performed under the minimum stress against animals. Because the owners desired simple screening test for gingiva and periodontal areas without sedation and/or anesthesia.
So, we did not evaluate the periodontitis. Of course, it was likely that some of the dogs had a periodontitis, I think so. But I could not evaluate exactly.
The diagnostic criteria of gingivitis were described in text.
Throughout the text: in vitro and in vivo italicized
>
Thank you. I corrected.
1. Introduction
normal flora - commensal microbiota
>
Thank you for indication. I corrected.
2. Materials and Methods
2.1. Ethics Statements and Sample Collection
The authors report that animals have gingivitis, but there are varying degrees of gingivitis. Reference to this assessment (gingival index) should be provided.
>
Thank you for suggestion.
I described the criteria of gingivitis in text.
2.3. Bacterial Production of VSCs and Determination of VSCs
The concentrations of suspensions of pure-cultured bacterial colonies in axenic PBS (please explain PBS meaning)
>
Thank you for your advice.
I changed as follows: in axenic phosphate-buffered saline (pH 7.4)
Discussion
"development of systemic diseases is a concern."
In my opinion the local expression and especially the systemic expression of periodontal disease, which is not just a concern but which is very well documented in man and dog, should be the subject of a paragraph in the introduction.
>
Thank you for your advice.
I agree with your opinion.
So, I removed this paragraph to introduction.
5. Conclusions
2.2) ?
>
Thank you for your indication.
It was my simple mistake.
So, I corrected to 2.2.
References
Despite being old and the evaluation method being different, I believe that this reference should be taken into account: Oral malodor measurements on a tooth surface of dogs with gingivitis. Hennet PR, Delille B, Davot JL.. Am J Vet Res. 1998Mar;59(3):255-7.
On the other hand, this article may allow the authors to enrich the text with the explanation of the role of these compounds in the pathophysiology of periodontal disease.
Role of Hydrogen Sulfide in Oral Disease. Wu DD, Ngowi EE,Zhai YK, Wang YZ, Khan NH, Kombo AF, Khattak S, Li T, Ji XY.Oxid Med Cell Longev. 2022 Jan 25;2022:1886277. doi:10.1155/2022/1886277. eCollection 2022.
>
Thank you for your advice.
I added these two articles in references.
Round 2
Reviewer 2 Report
I appreciate your efforts in improving the manuscript « Volatile Sulfur Compounds Produced by the Anaerobic Bacteria Porphyromonas spp. Isolated from the Oral Cavities of Dogs» (vetsci-2513181). There are a lot of improvements in the text and in the Table 2. The only problem is still there: the strange data of 64.2± 126.3 ppb, (of the dimethyl sulfide ((CH3)2S) production by Porphyromonas gulae). I wrote: “The authors must clarify this matter in the text”. The Authors wrote: “I inserted one sentence in text”, but this sentence was only about this fact without explanations. It will be nice if the authors can explain this fact. I understand that it is difficult and will not insist on this matter in this manuscript .

Author Response
Reviewer 2
Comments and Suggestions for Authors
There are a lot of improvements in the text and in the Table 2.
The only problem is still there: the strange data of 64.2± 126.3 ppb, (of the dimethyl sulfide ((CH3)2S) production by Porphyromonas gulae). I wrote: “The authors must clarify this matter in the text”. The Authors wrote: “I inserted one sentence in text”, but this sentence was only about this fact without explanations.
It will be nice if the authors can explain this fact. I understand that it is difficult and will not insist on this matter in this manuscript.
>
Thank you for your suggestion.
As you know, I cannot explain this matter now.
So, I deleted this sentence.
Reviewer 3 Report
Materials and Methods
of 25 mails (males)
The description of the Gingival index does not respect the reference and must be rewritten accordingly.
Results
Table 2 - Number of colonies instead examined numbers
Author Response
Reviewer 3
Comments and Suggestions for Authors
Materials and Methods
of 25 mails (males)
>
Thank you for your indication.
It was my simple mistake.
I corrected.
The description of the Gingival index does not respect the reference and must be rewritten accordingly.
>
Thank you for your indication.
I changed the description regarding gingival index, and changed the reference.
Results
Table 2 - Number of colonies instead examined numbers
>
Thank you for your suggestion.
I corrected.